

# RNA expression and disease tolerance are associated with a "keystone mutation" in the ochre sea star *Pisaster ochraceus*

V. Katelyn Chandler[1,†] and John P. Wares[1,2]

[1] Department of Genetics, University of Georgia, Athens, GA, United States of America
[2] Odum School of Ecology, University of Georgia, Athens, GA, United States of America
[†] Deceased.

## ABSTRACT

An overdominant mutation in an intron of the elongation factor 1-$\alpha$ (EF1A) gene in the sea star *Pisaster ochraceus* has shown itself to mediate tolerance to "sea star wasting disease", a pandemic that has significantly reduced sea star populations on the Pacific coast of North America. Here we use RNA sequencing of healthy individuals to identify differences in constitutive expression of gene regions that may help explain this tolerance phenotype. Our results show that individuals carrying this mutation have lower expression at a large contingent of gene regions. Individuals without this mutation also appear to have a greater cellular response to temperature stress, which has been implicated in the outbreak of sea star wasting disease. Given the ecological significance of *P. ochraceus*, these results may be useful in predicting the evolutionary and demographic future for Pacific intertidal communities.

## INTRODUCTION

The dynamics of an ecological community are often studied in terms of abundance and interactions of the component "macro" diversity - predation, competition, parasitism, even positive interactions among community members (*Duffy & Hay, 2001*; *Bruno, Stachowicz & Bertness, 2003*; *Vellend, 2016*). Yet it is also clear that pathogens play immense roles in the elimination of weaker or maladapted individuals, affecting density and behavior of infected organisms, and even reshaping a community (*Harvell et al., 2002*; *LoGiudice et al., 2003*; *Mitchell & Power, 2006*; *Stephens et al., 2016*). Despite many early advances in community ecology being derived from marine communities (*Connell, 1961*; *Paine, 1969*), our understanding of disease in marine organisms remains quite limited, with a few notable studies that have guided contemporary work (*Jolles et al., 2002*; *Lafferty, Porter & Ford, 2004*; *Mydlarz, Jones & Harvell, 2006*; *Sweet, Bulling & Cerrano, 2015*).

A recent and dramatic pandemic known as "sea star wasting disease" (SSWD) has led to very high mortality in a large number of sea star species on the Pacific coast of North America (*Hewson et al., 2014*; *Eisenlord et al., 2016*; *Menge et al., 2016*; *Montecino-Latorre et al., 2016*). By eliminating many predators from these coastal ecosystems, SSWD

Corresponding author
John P. Wares, jpwares@uga.edu

provides novel opportunities to evaluate hypotheses of how species can drive community composition (*Menge et al., 2016*; *Gravem & Morgan, 2017*) as well as further exploration of the pathogen or pathogens that may be driving these changes (*Hewson et al., 2014*). A component of understanding how marine communities will respond to disease in general involves exploration of host diversity for traits that influence susceptibility or mortality (*Vollmer & Kline, 2008*; *Wright et al., 2017*). Often there are evolutionary trade-offs that influence host diversity that are dependent on the prevalence of particular pathogens (*Aidoo et al., 2002*; *Gemmell & Slate, 2006*).

The sea star *Pisaster ochraceus*—best known as a "keystone predator" that modifies the diversity of its intertidal community (*Paine, 1969*)—harbors a mutation in the elongation factor 1-α (EF1A, hereafter) gene that is characterized as 'overdominant' (*Pankey & Wares, 2009*); that is, where heterozygous individuals (carrying one copy of this mutation) have dramatically higher fitness than either homozygote. At the time, with no apparent mechanism for this heterozygote advantage, *Pankey & Wares (2009)* noted that overdominance has often been associated with disease tolerance (*Gemmell & Slate, 2006*). Recent field surveys of apparently healthy and diseased individuals of *P. ochraceus* suggested that individuals carrying the insertion mutation (*ins*) described by *Pankey & Wares (2009)* have lower prevalence of (or mortality from) SSWD than individuals homozygous for the wild-type (*wild*) sequence (*Wares & Schiebelhut, 2016*).

The EF1A gene produces a "housekeeping" protein that is involved in translational elongation (forming peptide bonds between amino acids) of newly-generated proteins. However, EF1A also appears to be involved in diverse cellular functions (*Ejiri, 2002*), and diversity at this gene has been implicated in variation in fitness in other metazoans (*Stearns, 1993*; *Stearns & Kaiser, 1993*). Currently, the mechanism by which the *ins* mutation— which is within an intron between two coding subunits (*Pankey & Wares, 2009*)—affects the function of EF1A or the cellular functions associated with SSWD (*Hewson et al., 2014*) remains unknown, and of course the *ins* marker may simply be linked to another polymorphism that is actually promoting these effects. However, the "moonlighting" functions of EF1A (*Ejiri, 2002*) include mediating responses to viral infection (*Li et al., 2013*; *Wei et al., 2014*) and inflammation (*Schulz et al., 2014*), as well as environmental stress (*Bukovnic et al., 2009*). It is not unusual for stress to be cited as a component of disease susceptibility (*Cohen et al., 2012*). Current evidence is mixed about what environmental stressors have promoted SSWD in *P. ochraceus*, with some studies suggesting that elevated water temperature (*Bates, Hilton & Harley, 2009*; *Eisenlord et al., 2016*) influenced the outbreak in one region (the Salish Sea) and another study on the Oregon coast indicating the opposite, that cooler water from upwelling may have been a physiological stressor (*Menge et al., 2016*).

We can now query distinct genotypes for variation in RNA transcription to identify components of cellular and molecular networks that are associated with specific trait variation (*Cohen et al., 2010*). Here we test two hypotheses using RNA sequencing of a set of individuals of each EF1A genotype in *P. ochraceus* (the mutation is homozygous lethal, so there are only two genotypes for this marker). First, the mutation (or linked diversity) could influence the overall regulation of other genes, in which case we may

detect significantly different expression of a set of loci between *ins* heterozygotes and *wild* homozygotes. Second, we evaluate how individuals of each genotype respond to thermal stress, as the effects of many mutations will be environment-dependent (*Rutter et al., 2017*). Temperature shifts are associated with changes in feeding (*Sanford, 1999*), metabolism (*Fly et al., 2012*), and intertidal distribution (*Menge et al., 2016*) in *P. ochraceus*. Thus, we coupled a temperature challenge trial with behavioral observations and repeated RNA sequencing to understand how individuals respond to periods of elevated temperature or stress. In this case, we hypothesized that an interaction between environmental stress and cellular physiology could be indicated by distinct patterns of activity levels or changes in expression across the two *ins* genotypes.

Our goal is to illuminate mechanisms by which EF1A *ins* heterozygotes in *P. ochraceus* may be protected from SSWD, as this information may guide exploration of why some sea stars in the Pacific intertidal community are more susceptible than others to this disease. Additionally, this system provides an opportunity to explore how variation in expression of a gene or gene network that is of fundamental importance to organismal development, growth, and acclimation can affect the tolerance of an organism to disease.

## METHODS

### Field and lab

Individual *P. ochraceus* were collected from ∼0 m tidal depth within the Friday Harbor Laboratories marine reserve (Friday Harbor, WA, 48.54°N 123.01°W) in June 2016. Collections were made following written permission from the Associate Director of the Friday Harbor Laboratories. Individuals were placed in sea tables with ambient temperature, unfiltered, running sea water within 1 h of collection and fed available bivalves *ad libitum*. After the experiment, all surviving individuals were returned to the field.

At the beginning of the experiment two samples (∼25 mg) of tube feet were removed from each individual; one sample was placed in 95% undenatured ethanol (for genotyping as in *Wares & Schiebelhut, 2016*), the other sample into RNALater (Thermo Fisher). Tissue sampling was repeated following the heat trial described below. Distal tube feet were used in part to minimize damage to individual *P. ochraceus*, and to standardize contrasts of regulatory change (*Montgomery & Mank, 2016*). Individuals were kept in flow-through sea tables in Vexar enclosures to ensure consistent individual identification. DNA samples were tested for presence of SSaDV (the putative pathogen causing SSWD) using qPCR as in *Hewson et al. (2014)*.

Righting responses (Fig. 1) were used to explore the physiological status of individuals of each genotype subjected to periods of elevated temperature. Increasing the temperature by ∼3° is known to influence the physiology of *P. ochraceus* (*Sanford, 1999*; *Fly et al., 2012*). Flow-through temperature treatments were performed as in *Eisenlord et al. (2016)*; individuals were maintained at +3 °C for 8 days. Sea table temperature was monitored 4× daily with digital thermometers and with Hobo Tidbit data loggers. Righting response trials were performed as in *Held & Harley (2009)*. We recorded the time each individual required to flip from the aboral side to the point that the majority of arms contacted the
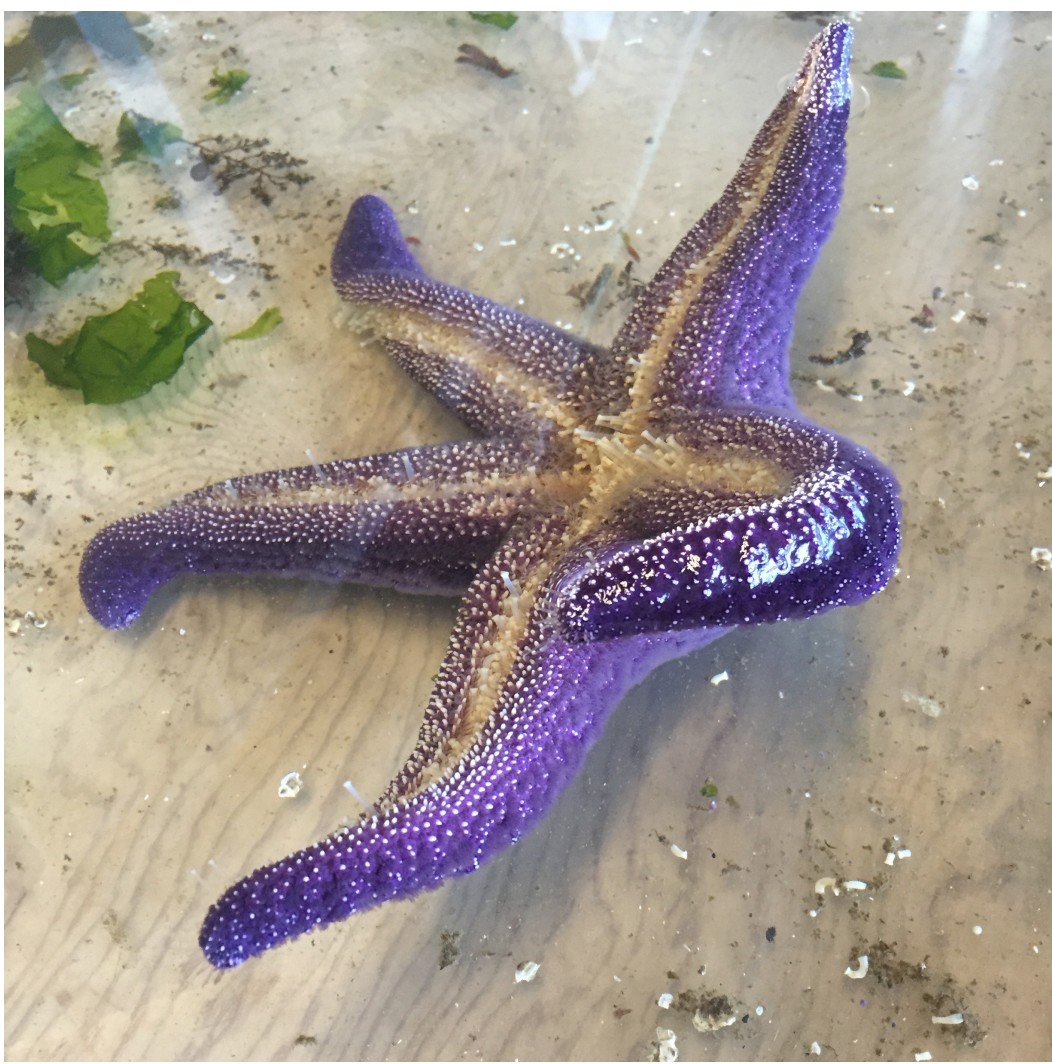

**Figure 1** **Image of *Pisaster ochraceus*.** Aboral view of *Pisaster* in righting response trial. Photo by JPW.

surface on their oral side. Trials were performed three times in each condition: in ambient seawater, at the end of the temperature trial, and again when individuals returned to ambient temperature. Individuals that did not right themselves within 1 h were considered unresponsive and were excluded from subsequent analyses. Minimum and mean righting response times were recorded; these values are examined across EF1A genotypes using a *t*-test as well as a linear mixed-effects model evaluating response to temperature by genotype using the *lmerTest* package (*Kuznetsova, Bruun Brockhoff & Haubo Bojesen Christensen, 2016*) in R version 3.3.2 (*R Core Team, 2016*).

## RNA sequencing and comparison

Samples of tube feet stored in RNALater were thawed on ice and 25 mg were removed for RNA isolation using a Qiagen RNEasy Mini-prep kit. A Qiagen TissueRuptor with sterile disposable pestles was used for homogenization of each sample. RNA samples were

submitted to the Georgia Genomics Facility (GGF; dna.uga.edu) for stranded RNA library preparation (Illumina TruSeq LT) and subsequent quality checks using an Agilent 2100 BioAnalyzer. Libraries were sequenced in parallel (high output PE75) on an Illumina NextSeq 500 at GGF and then informatically demultiplexed.

Our pipeline followed *Kelly et al. (2017)*, both with and without the utilization of *cd-hit* (*Li & Godzik, 2006*) to reduce the sequence complexity in the data, using a sequence similarity threshold of 98%. Illumina adapter sequences were removed during the demultiplex step. FASTQ data were cleaned using Trimmomatic (*Bolger, Lohse & Usadel, 2014*) (default settings), and two transcriptome assemblies were generated using *in silico* read normalization in Trinity (*Grabherr et al., 2011*). The first assembly utilized data from all 20 RNA libraries; the second utilized only the data from four individuals, two of each genotype, chosen for high RIN values and read numbers. Trinity *de novo* assembly was performed on a Georgia Advanced Computing Resource Center 512 GB node with eight processors. Individual RNA libraries were then aligned to the assemblies using Bowtie2 (*Langmead & Salzberg, 2012*) and the RSEM method (*Li & Dewey, 2011*) as in *Haas et al. (2013)*.

All assembled Trinity clusters were used as *blastx* queries against the *nr* database, restricted to GI numbers for Echinodermata, with the best hit for each (*e*-value $< 10^{-6}$) retained. A custom R script was used to collapse the expression count files by inferred gene and by BLAST homologies except where otherwise noted. Differential expression was quantified using edgeR (*McCarthy, Chen & Smyth, 2012*), filtering reads for a counts-per-million (CPM) $> 1$ in at least two of the libraries. Other filtering combinations were attempted with similar results (VK Chandler, results not shown). Both negative binomial and empirical Bayes dispersion measures were estimated before testing for differences. The libraries representing ambient and elevated temperature exposure were evaluated individually by genotype for differential expression between treatments, as well as a paired sample analysis using edgeR.

Additionally, a sorted and unsorted permutation test of genotype contrasts was performed to ensure that the EF1A genotype explained the greatest pattern of differentiation among these samples. The sorted permutation test evaluated the number of differentially expressed genes between the two genotype classes against a distribution generated from (a) moving one library at a time into the other classification, (b) all permutations in which one from each classification is moved to the other, and (c) where two libraries from each classification are moved to the other. These were repeated for all such possible permutations. The unsorted permutation test randomly drew libraries without replacement to comprise two classes of equal size and repeated the contrast between actual number of differentially expressed genes and the permutational distribution of this value.

To specifically consider differential expression of EF1A, we considered all fragments that successfully BLAST to NCBI accession AB070232, a ~5 kb sequence of the EF1A gene region from the confamilial *Asterias amurensis* (*Wada et al., 2002*), and also used sequence data (NCBI KY489762–KY489768) generated from cloning of *P. ochraceus* EF1A (*Pankey & Wares, 2009*) to identify any expression of the focal intron region that harbors the *ins* mutation. These latter assemblies were performed using Geneious R10 (Biomatters).

## RESULTS

A total of 24 individuals were collected from the Friday Harbor Laboratories marine reserve, 1 was returned due to injury, and 20 survived our lab trials (one individual, an *ins* heterozygote, died of apparent SSWD; 2 others from distinct external infections) and were returned to their original location. As in previous studies (*Pankey & Wares, 2009*), the ratio of heterozygotes (+/*ins*, or *ins*) to homozygotes (+/+ or *wild*) at the EF1A locus was ~1:1. In order of initial labeling, the first 5 individuals of each genotype that had complete behavioral data were selected for RNA sequencing (Table S1). Each individual was genotyped 3 times from 3 separate tissue samples with no errors. These 10 individuals exhibited no visible signs of SSWD and tested negative for SSaDV.

### Behavior

For the 10 individuals analyzed in full, righting response trials (Fig. 1) suggested that *ins* heterozygotes righted themselves approximately 1.8 times faster than *wild* homozygotes (Table S1; $t$-test $p = 0.01$; linear model $p = 0.022$) at both temperatures. However, including all data on righting response (from all 17 individuals with complete behavior data; $n = 6$ unresponsive individuals were evenly distributed across genotypes) introduces higher variation in response by genotype; the effect is in the same direction but not significant ($t$-test $p = 0.77$; linear model $p = 0.737$). Including *wild* individual "Po5" in the *ins* genotypic class (see below for rationale) strengthens these results but they are still not statistically significant when all individuals are included.

### Sequenced RNA diversity

Table S1 provides information for each library used in transcriptome assemblies; all sequence data are available from NCBI (BioProject PRJNA357374). Of the two *de novo* assemblies, the reduced-input transcriptome had greater length and quality of contigs ($N_{50}$ of 1,799 bp, median contig length 513, total assembled bases 179,034,265) and is the focus of subsequent analyses. Fragments that were differentially expressed (False Discovery Rate (FDR) <0.01) between the two genotypes from the two Trinity assemblies were themselves *de novo* aligned in Geneious R10; 80.76% of contigs from one of the two assemblies aligned with one from the other. Using a 98% threshold with *cd-hit* reduced the total number of sequences from 179,563 to 154,150. All results are qualitatively similar across all three assemblies, but the results reported hereafter are based on the *cd-hit* reduced assembly.

The total of 154,150 transcripts analyzed represented 62,713 Trinity clusters and 110,525 gene regions after isoforms were summed for each. Additionally, expression counts were summed for fragments with identical NCBI gi numbers, reducing the total number of expressed fragments to 107,189. A total of 9,953 distinct gi values were recovered from *blastx* ($e$-value $< 10^{-6}$; 9,563 of these were hits to *Strongylocentrotus purpuratus*). Table S1 shows information for all fragments that passed our CPM filters.

#### Permutational testing

For each iteration of the sorted permutation test, the number of gene fragments that are significantly different (FDR $< 0.01$) was identified and contrasted with the true

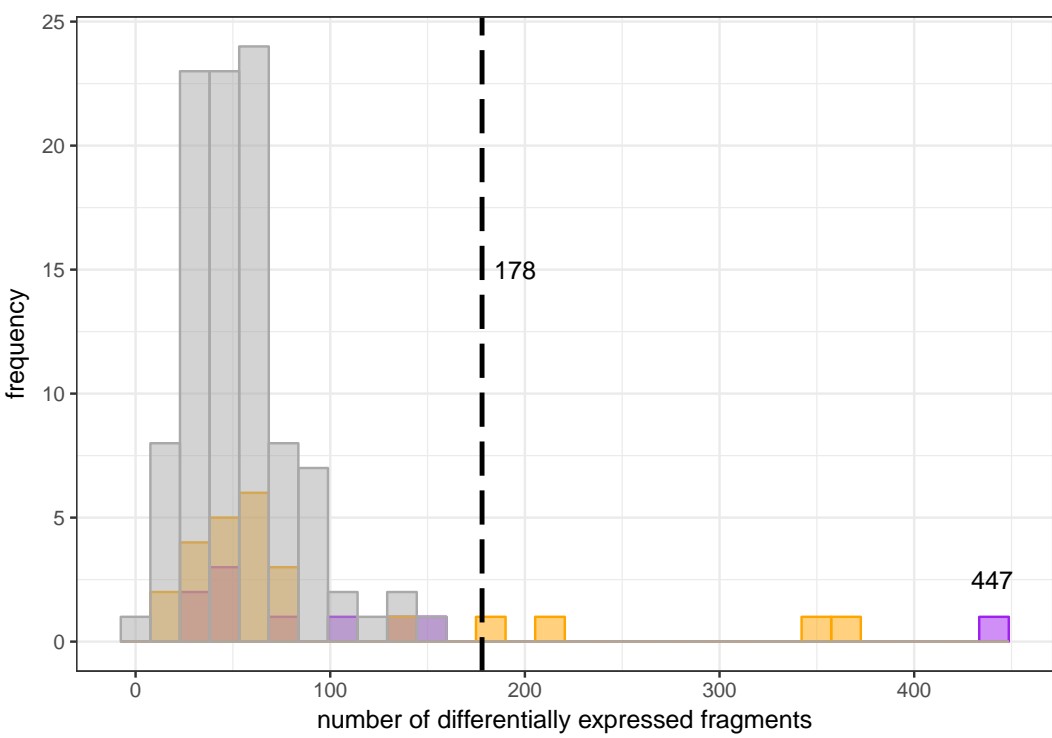

**Figure 2   Randomized expression differences among libraries.** Sorted permutational misassignment of individuals and comparison with actual EF1A genotype partitions. Misassignments were directed to maintain (nearly) equal sample sizes in the two groups. Histogram bars in purple indicate reassignment of a single library; orange represents reciprocal swap of single libraries across partitions. Grey represents reciprocal swap of two libraries across genotype partitions. Partitioning of individuals by EF1A genotype suggests a stronger signal (vertical dotted line) than almost all permutations; 'misassignments' with more extreme results always involve reassignment of wild individual Po5. The maximal observation shown (447 differentially expressed fragments) represents Po5 as an ins heterozygote instead.

classification. The results suggest that differentiation of the two genotypes is robust relative to the most extreme misassignments (Fig. 2), and greater than 0.96 of all permutations. All permutations with higher counts of differentially expressed transcripts involve reassignment of individual Po5 (*wild*); though EF1A genotype was confirmed for this individual, it is similar to the *ins* heterozygotes for many expression traits (Table S1). Full unsorted permutation testing with 500 permutations also showed that the effect size using EF1A genotypes as a means of partitioning the data is large relative to random (96.8th percentile). If library Po5 is excluded, the expression differences between genotype classes is greater than any permuted re-sampling of the data.

### Comparison of differential expression across genotypes

There are strong differences in the constitutive expression patterns of the 5 *wild* and 5 *ins* individuals assayed. There are 178 fragments exhibiting differential expression with FDR < 0.01, and 18 with FDR < 0.0001 (Fig. 3). As above, if library Po5 is excluded, a greater number ($n = 395$) of fragments exhibit differential expression (FDR < 0.01), suggesting that this individual represents an inconsistent expression phenotype for its

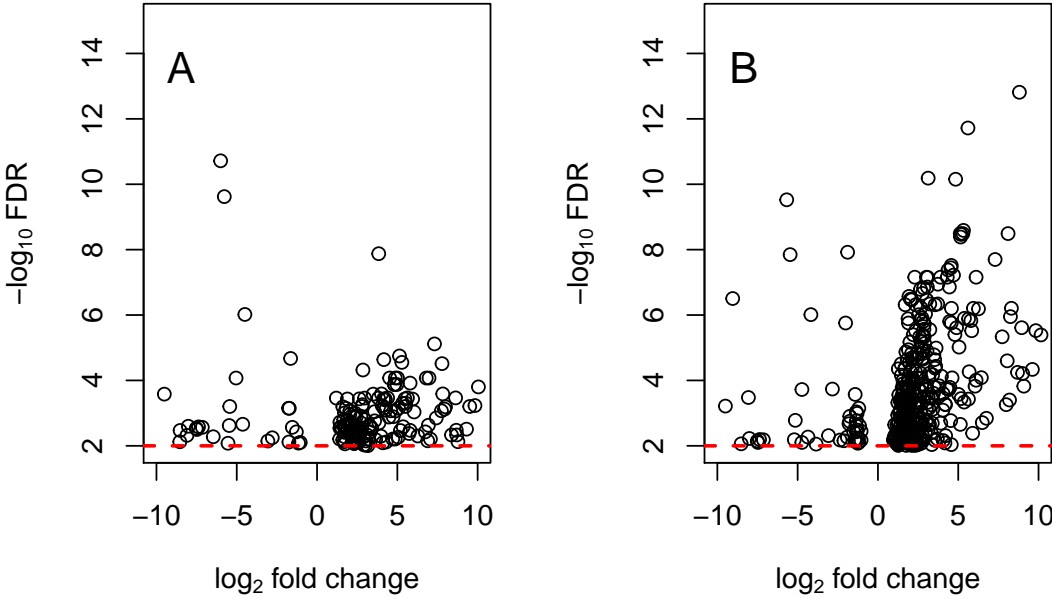

**Figure 3** **Differential expression of EF1A genotypes.** Volcano plot of transcripts that are differentially expressed between EF1A genotypes (FDR < 0.01). Contrast indicated with positive logFC values on the right for fragments that have higher expression in *wild* homozygotes. (A) includes all individuals in study; right panel (B) excludes *wild* individual Po5. Red dotted lines indicate FDR of 0.01.

EF1A genotype (see heatmaps in Table S1). If library Po5 is instead categorized as an *ins* heterozygote, there are 447 distinct transcripts between the two genotypes (FDR < 0.01). The full paired analysis (using all 10 libraries from both temperature treatments) identifies a similar number of DE loci overall, and a similar differentiation is identified between genotypes at elevated temperatures.

The effect of the *ins* genotype appears to be inhibitory; Figure 3A shows only those fragments that are differentially expressed between the two genotypes, and only 25 of 178 fragments with FDR < 0.01 exhibit higher expression in heterozygotes. Many of the significantly elevated transcripts in heterozygotes are modestly expressed compared to the significantly elevated transcripts from wild homozygotes. The average log CPM for fragments with FDR < 0.01 that are more highly expressed in *ins* heterozygotes is 0.876 (maximum 6.154), while the same average for fragments that are more highly expressed in *wild* homozygotes is 3.639 (maximum 11.217). A similar, but stronger, result is obtained when Po5 is excluded, with only 45 of 395 differentially expressed (FDR < 0.01) fragments being more highly expressed in heterozygotes (Fig. 3B); for differential expression at the level of FDR < $10^{-4}$, only 6 of 131 fragments are more highly expressed in *ins* heterozygotes.

### Response to elevated water temperature

Following exposure to water warmed by +3 °C, *wild* homozygotes exhibited a larger number of potential loci ($n = 46$) that changed in expression (FDR < 0.01) than *ins* heterozygotes ($n = 6$; see Table S1). Using a more inclusive cutoff (FDR < 0.1, as in *Wright et al., 2017*) amplifies this contrast, with *wild* homozygotes showing differential expression at 197 regions and heterozygotes at only 14 gene regions. If individual Po5
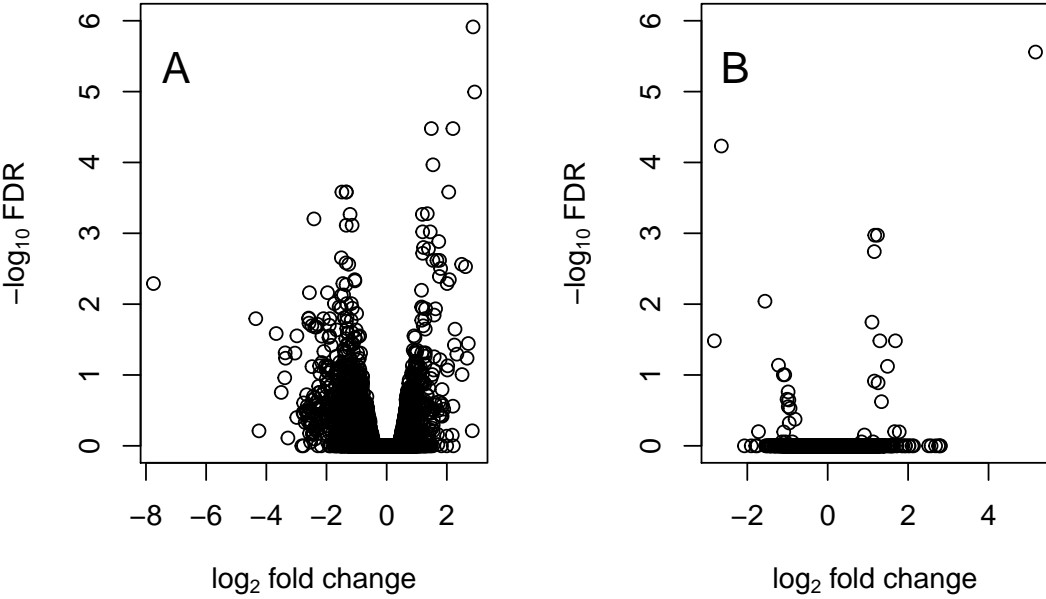

**Figure 4 Differential expression after heat exposure.** Reaction by genotype to increased water temperature in *P. ochraceus*. (A) 46 fragments exhibit significant (FDR < 0.01) differential expression between ambient and elevated temperatures in *wild* homozygotes. (B) 6 fragments exhibit significant (FDR < 0.01) differential expression after temperature treatment in *ins* heterozygotes.

is excluded, the remaining *wild* homozygotes then exhibit 62 fragments that change in expression (FDR < 0.01; 190 with FDR < 0.1), suggesting again that the expression phenotype of this individual adds considerable variance to the expression patterns of homozygotes. Additionally, the average effect of temperature exposure appears to be in opposite directions: while *ins* individuals have a mean log fold change in expression across loci of $0.278+/-2.132$, *wild* homozygotes have a mean reduced expression across loci (log fold change $= -0.455+/-1.685$). Of all fragments identified as responding to the temperature treatment, 3 of 6 identified in the heterozygotes are also found among those that are differentially expressed in the homozygotes (whether or not Po5 is included, at FDR < 0.01). These results are suggestive that homozygous individuals experienced a greater net change in expression phenotype following exposure to heat than *ins* heterozygotes (Fig. 4).

### Elongation factor 1-alpha

Following BLAST analysis, only 1 fragment sufficiently matched NCBI accession AB070232 (*Wada et al., 2002*), a ~5 kb sequence of EF1A from *Asterias amurensis*. This fragment does not appear to be differentially expressed (FDR < 0.01) between *wild* and *ins* EF1A genotypes. Including other fragments that have sufficient homology to "elongation factor 1-α" (a partial fragment from *Patiria miniata*) provides similar results, whether or not individual Po5 is included. Previous analyses that assessed BLAST homology using the *blastn* algorithm against *A. amurensis* also showed that summing across all putative EF1A homologs indicated no significant expression differences at this locus (DOI:10.7287/peerj.preprints.2990v1). Assembly of RNA sequence fragments from libraries

of the two genotypes to the full *A. amurensis* EF1A sequence showed no obvious distinctions in coverage of coding regions (results not shown).

## DISCUSSION

The data and results presented here lead to a remarkable conclusion—that the canonical intertidal predator, *Pisaster ochraceus*, after decades of intensive ecological scrutiny, appears to include two physiologically distinct types. We know that these types are not reproductively isolated (*Pankey & Wares, 2009*), so these distinct types are formed each generation via an overdominant polymorphism that influences regulation of gene expression. The two forms significantly (FDR < 0.01) differ at 0.404% of all expressed fragments analyzed here, a small but important proportion (Figs. 2 and 3). These data are consistent with a mutation in a regulatory region in that *ins* heterozygotes have a limited expression of many of the differentially expressed loci relative to *wild* individuals (Fig. 3). Perhaps more importantly, the two forms responded very distinctly to temperature stress (Fig. 4), with a qualitatively distinct expression change profile for *wild* homozygotes than *ins* heterozygotes. The results from one unusual individual (Po5) clearly suggest, however, that the *ins* genotype marker used to separate these groups may not be the causal mutation for these cellular and physiological shifts. *Pankey & Wares (2009)* had discussed linked polymorphic diversity using cloned and sequenced fragments of the EF1A intron that carries the mutation, but expressed concerns about PCR-mediated recombination in those data. It now seems likely that Po5 harbored a recombination event, appearing to be *wild* but with the expression profile of an *ins* individual. Future work will explore this region in greater detail.

Intriguingly, there is now a linkage between these physiological changes that seem to reduce the individual response to temperature stress and the *ins* marker that is associated with reduced incidence or mortality to SSWD (*Wares & Schiebelhut, 2016*). *Harvell et al. (2002)* noted that a warming climate could affect the development or survival of pathogens, but certainly could also interact with host physiological stress as well (*Cohen et al., 2012*). Our study is limited in understanding this linkage in several ways—there has been little or no experimental annotation of differentially expressed genes in *P. ochraceus*, and our temperature contrast experiment did not include control replicates to assess the effect of stress from being held in our mesocosms with or without temperature manipulation. Nevertheless, the effect of our temperature stress trial suggests heritable variation in how individuals respond to heat stress. As elevated temperatures may accelerate SSWD (*Bates, Hilton & Harley, 2009*; *Eisenlord et al., 2016*; but see *Menge et al., 2016*), the likelihood that the mutation linked to the *ins* marker ameliorates multiple forms of stressors on the health of an individual is worth further investigation.

As *P. ochraceus* is likely to vary behavior along with physiological stress (*Monaco et al., 2015*), we also evaluated whether there was an interaction between the EF1A genotype, temperature stress, and behavioral activity. Righting responses were used to understand the response to heat as an influence on activity levels (*Held & Harley, 2009*). Heterozygous individuals tend to right themselves more quickly in a limited sample. However, individual-level variation was high and the biological effect of genotype on this response may be low

or absent. Individuals appeared to be consistent in their response, *i.e.,* individuals with long response times tended to do so at all treatments; whether this is associated in any way with effects of this genotype requires further consideration. Overall, we conclude that righting response is a noisy response variable and perhaps ineffective for assaying physiological contrasts. We are not the first to recognize this difficulty:

> "It could probably be said, in a word, that the starfish may, and does, in different cases, right itself in any conceivable way, - and indeed, in many ways that would not readily be conceived before they were observed."

> *Jennings (1907)*

Thus, other approaches such as respirometry (*Fly et al., 2012*) are needed to more directly understand the genetic basis of stress response in *P. ochraceus.*

It is also notable that the regulatory effect of the *ins* mutation (or a linked polymorphism) has a consistent response - there is a clear asymmetry (Fig. 3) in expression of transcripts suggesting that the *ins* mutation affects a promoter region. The genomic features that are differentially expressed in response to the *ins* mutation are of interest, but accurate functional annotation of these transcripts are currently limited by the tremendous evolutionary divergence between *Pisaster* and other characterized Asteroid genomes (*Patiria miniata* (echinobase.org) and *Acanthaster planci* (*Hall et al., 2017*)). Generating a more extensive list of loci that are coregulated by the *ins* marker is of modest utility without better experimental data in this non-model organism (*Hudson, Dalrymple & Reverter, 2012*). We do not know if the differentially expressed loci are relatively rapidly evolving, or if these transcripts represent noncoding RNA; currently, these hypotheses are difficult to test with available resources (*Dinger et al., 2008*). Our ability to explore the effects of differential genotype in *P. ochraceus* may also require an understanding of tissue specificity. Here, tube feet were used as simple non-invasive tissues for sampling because the health of the local population is of concern. Future efforts could target tissues more specific to immune response function. For example, EF1A is already thought to regulate interleukins (*Schulz et al., 2014*), which are represented among the identifiable differentially expressed gene regions in this study (Table S1). These are thought to be produced in the axial organ and are a basic component of the echinoderm immune response (*Mydlarz, Jones & Harvell, 2006*; *Leclerc & Otten, 2013*) that stimulate coelomocytes and are associated with antiviral activity (*Ghiasi et al., 2002*).

A polymorphism like this should not be stable unless there is some balance of benefits to both genotypes (*Subramaniam & Rausher, 2000*). With typical genotype frequencies in the wild (*Pankey & Wares, 2009*; *Wares & Schiebelhut, 2016*), approximately 1/16th of all offspring (1/4 of the offspring from 1/4 of the random mating events) are lost each generation to this polymorphism. Similar levels of reduced fitness are involved in explorations of Dobzhansky-Muller interactions associated with outbreeding depression (*Sweigart, Fishman & Willis, 2006*). This is a considerable mutational load attributed to a single polymorphism yet the sudden appearance, or incidence in recent decades, of high mortality events like SSWD is unlikely to be a sufficient mechanism for maintaining this

polymorphism. The two allelic classes each harbor considerable levels of flanking diversity and appear to be relatively divergent and ancient (*Pankey & Wares, 2009*), and the high frequency of the *ins* allele throughout the range of *P. ochraceus* (*Pankey & Wares, 2009*; *Wares & Schiebelhut, 2016*) suggests its origin is not recent (*Slatkin & Rannala, 2000*).

The question remains, what has maintained this polymorphism, and what can we learn from this about disease in other echinoderms—or more broadly, other animals—about interactions of stress and pathogens? In other major epidemics, it has been noted that mortality has been highest in individuals that are weak *or* that have the strongest inflammatory/immune response to a pathogen (*Lai, Ng & Cheng, 2015*). If this is true, perhaps *wild* individuals are more prone to extreme stress responses. The reality is that stress tolerance is thought to be highly context-dependent (*Berry et al., 2011*; *Bay & Palumbi, 2015*) and may be difficult to assess in a wild population such as the *Pisaster* surveyed here. Each individual bears high levels of additional variation that mediates their responses to environment, pathogens, and so on. The fact that the *ins* mutation is associated with such strong biological effect amidst the noise of other natural genomic diversity is extraordinary. Certainly there are other examples of single mutations that confer significant health and life history consequences on carriers (*Aidoo et al., 2002*; *Drnevich et al., 2004*; *Gemmell & Slate, 2006*). Distinct phenotype classes within a species often have distinct expression profiles (*McDonald et al., 1977*; *Garg et al., 2016*), including instances of disease or tolerance phenotypes (*Emilsson et al., 2008*; *O'Connor et al., 2017*). Our hopes are that further exploration of this system, in an ecologically important sea star (*Paine, 1969*; *Menge et al., 2016*), will be of relevance for a more general understanding of health and pathogen tolerance.

## IN MEMORIAM

Virginia Katelyn Chandler unexpectedly passed away in June 2017 after submission of the first version of this manuscript; her contributions to this project, in just her first two years at The University of Georgia, were outstanding. Deepest condolences to her family and friends, from all in the Wares Lab and at the University of Georgia.

## ACKNOWLEDGEMENTS

We thank M Kelly, J Griffiths, A Del Rio, M Bitter, K Coombs, J Wong, J Wallace, M Mintz, and R Wares for field and lab efforts that contributed to this work at Friday Harbor Laboratories (FHL). Additionally, support from M Eisenlord, B Swalla, M Dethier, S George, J Hodin, and the staff at FHL was invaluable. Technical input from D Barshis, S Caplins, K Bockrath, M MacManes, K Thornton, and S Pankey was greatly appreciated. Computational support, particularly D Brown, Y Huang, R Masalia, and A Bewick along with lab support by R Nilsen, was greatly appreciated. Ideas generated in discussion with S Gravem, L Schiebelhut, and M Dawson helped develop this project. S Pankey, S Heisel, J Hamlin, I Hewson, K Bockrath, E Sotka, F Barreto, and C Ewers-Saucedo evaluated early drafts of the manuscript; E Sotka, M Dawson, S Pankey, and M Kelly helped with later versions along with M Lloyd and one anonymous reviewer.

### Funding

This work was supported in part by the National Science Foundation (Ecology of Infectious Diseases 1015342) and the University of Georgia Research Foundation. There was no additional external funding received for this study. The funders had no role in study design, data collection and analysis, decision to publish, or preparation of the manuscript.

### Grant Disclosures

The following grant information was disclosed by the authors:
National Science Foundation.
University of Georgia Research Foundation.

### Competing Interests

The authors declare there are no competing interests.

### Author Contributions

- V. Katelyn Chandler performed the experiments, analyzed the data, prepared figures and/or tables, reviewed drafts of the paper.
- John P. Wares conceived and designed the experiments, performed the experiments, analyzed the data, contributed reagents/materials/analysis tools, wrote the paper, prepared figures and/or tables, reviewed drafts of the paper.

### Field Study Permissions

The following information was supplied relating to field study approvals (i.e., approving body and any reference numbers):

Collection of animals and tissues was approved by the Associate Director of the Friday Harbor Laboratories in writing.

### DNA Deposition

The following information was supplied regarding the deposition of DNA sequences:

All sequence data are archived at NCBI in BioProject PRJNA357374.

### Supplemental Information

Supplemental information for this article can be found online at http://dx.doi.org/10.7717/peerj.3696#supplemental-information.

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
