# Peer review of "RNA expression and disease tolerance are associated with a “keystone mutation” in the ochre sea star Pisaster ochraceus"

_PeerJ, doi:10.7717/peerj.3696_

## Round 0.1 · original submission · Major Revisions

Thank you for submitting your work to PeerJ

Two reviewers have provided very thorough reviews and while they find the study interesting both suggest major revisions.

I agree with the points raised by both reviewers and would encourage you to resubmit a revised manuscript along with a point-by-point response to their comments as well as a description of how you addressed these points in the revised manuscript.

In your revision please pay particular attention to describing the limitations (biological and statistical) of the RNA-seq analysis and use those limitations to guide your interpretation, avoiding overly speculative conclusions. This will allow the reader to better contextualize the data and build upon your observations.

Please make it very clear where the RNAseq data has been deposited and if it is being used in subsequent analyses, either in progress or submitted.

The comment regarding the applicability of RNASeq to address splice variants of the EF1A gene is correct and could be addressed using isoform-specific PCR.

Given the scale and scope of the work, I agree with the reviewers that the overly speculation sections should be limited and the emphasis instead placed on how (and what kinds) of further experiments could provide additional insights.

I look forward to seeing a suitably revised manuscript.

Best,
Corey Nislow

Reviewer 1 ·

Basic reporting

Introduction:
The authors did a good job of explaining the history of the EF1A gene, and why this is the gene that they are focusing on. However, in the study the measure other things including "righting behavior" and heat stress, but these two things and why they are used, were not explained in the beginning, so it was hard for a non-echinoderm scientist to understand why these were included. Please change the text to reflect this, and to give more information on why you did what you did.

Figures need to flipped so that they can read easily. This may just be an issue of the uploading process.


The results do address the hypothesis that the authors set out to test. However, I am not certain that the results gained can be considered significant because of the low sample size and lack of replication of this experiment.

Experimental design

Methods:
The authors used RNAseq to analyze the expression of the EF1A gene, and compare amongst ins versus wild type.

Was this part of a larger overall gene expression study? Why were not other genes looked at? Since you sequencing the transcriptome, where is that information? If it is in another paper, please cite it, if not, than please include it in this paper. An overall expression analysis would be very helpful to set the context.

Under RNA Sequencing and Comparison:
Lines 117: examine what cd-hit is, and why you didn't use it.
Line 121: cross out "whole"


The sample size is very low, can you explain why this is so?

Also sample Po5 is mentioned alot because it was abnormal and taken out. However, it may have had biological meaning, so are there any ways to explore this individual more? It is hard for me to make a judgement because the experiment had such low replication.

Validity of the findings

Results:
Righting behavior: no difference

Hypothesis 1: EFIA had no expression differences across genotypes.

Hypothesis 2: constitutive expression differences between 5 wild and 5 ins.
Question: in the text please clarify how this went from 20 to now only analyzing 10 individuals.

Why were only 5 fragments identifiable? Did you confirm these with PCR to make sure that you are getting the whole region?

Heat exposure: homozygous had greater net change in expression due to heat stress compared to ins heterozygotes.

My worry with this result is that all of the statistical power is lost when you do a multidimensional scaling for all libraries, indicating that the results above are not accurate...and could be an artifact of the scaling. Can you please clarify this.

Again with Po5, why is individual so odd, and how did it's expression compare to others, is this an artifact of sequencing or truly a biological phenotype?

Line 232: please provide in supplementary data the multidimensional scaling plot of all libraries.

Lastly, how do you not know that the effects that you seeing the regulation of EFIA is not affected by other genes or mutations through out the genome? This needs to tested and examined more closely.

Additional comments

I think that this paper has a ton of potential and is interesting. Please just address the points I have stated, and lay out what is known for others who are not directly in this field. Also, including an analysis of the whole transcriptome will be very valuable and will add to what you are doing. Also, a better explanation of how you know that this mutation is the most important one, and not influenced by other genes in the genome.

Many of the conclusions at the end of the paper are extremely speculative, and because of the low number of individuals, I think that you cannot make any broad conclusions on stable polymorphisms etc.

·

Basic reporting

In my opinion, this article is well structured, well written, and presents a nice overview or relevant literature. I only have some very small suggestions:

Line 78: When were these animals collected?
Line 130: did you blastn against the nucleotide database or blastx against the protein database? It’s unclear here
Line 147: I’m very curious! What was the genotype of the 1 star that got SSWD?
Line 152: Did all 24 individuals test negative for SSaDV?
Line 165: More information about assembled transcriptome: number of transcripts, number of genes, % of genes with BLAST match?
Be more clear that the mutation is in an intron in the introduction. I see it is mentioned there but I didn’t really process that information until it is discussed on line 256
Add more info about transcript wide DE contigs: open reading frames? size? what percent of total transcript fragments was differentially expressed?
294-295: “Though a small number of fragments have sufficient BLAST homology to identified proteins (Supplementary Information S1)”. S1 doesn’t contain this information, as far as I can see.

Experimental design

I liked how the authors presented two hypotheses in the introduction and addressed each hypothesis in the results. While I think the RNASeq method is appropriate for addressing the second hypothesis (that the EF1A mutation is influencing the expression of other genes) is appropriate, I don’t think RNASeq is the best method for addressing the first hypothesis.
The first hypothesis, as written on lines 52-55 was:
“…the ins mutation – which is within an intron between two coding subunits of the EF1A gene (Pankey & Wares, 2009) –could affect mRNA splicing and thus generate subfunctional or functionally distinct transcripts.”
The RNASeq data presented here does not address possible splice variants of the EF1A gene arising from the ins mutation. I think a better way to address this question is to PCR the EF1A gene from cDNA to identify splice variants, clone them all, and Sanger sequence them. I think it would also be really helpful to be able to identify which transcripts arose from the wild type sequence and which (if any) arose from the mutation sequence in heterozygotes. Since the mutation is in an intron, I’m not really sure if that is possible without identifying another unique sequence differentiation between wild-type and mutant in the coding region. I also think some RT-qPCR might be helpful in being able to make clearer conclusions about expression differences between wild-type and mutation individuals.

I think the results presented by the authors that relate to the second hypothesis are sufficient for publication, but I don’t suggest publication of this article as it is currently written with both hypotheses.

Validity of the findings

I think the conclusions from the EF1A expression data are interesting but perhaps oversold since there isn’t an overall DE of all EF1A transcripts between wild-type and mutant. For example:
Line 252-255: “Homozygous wild individuals express some EF1A-like elements approximately twice as much as ins heterozygotes, suggesting that ins/ins homozygotes may not express this isoform or copy at all, perhaps the cause of early mortality (Pankey & Wares, 2009).” The EFI1-like elements being discussed here are not necessarily a unique isoform or gene copy, they’re just transcriptome fragments that all happened to be significantly differentially expressed. However, the overall expression of all EF1A fragments was not significantly different. It’s difficult to believe this story without knowing whether or not there are indeed unique isoforms of this gene.
I also feel Lines 269-290 oversell this story.

I find the transcriptome wide DE analysis to be the most interesting result here! I was wondering if it might be possible to eliminate some noise from the data to try to find more DE genes with BLAST hits. Have you tried filtering the transcriptome assembly with a program like Transdecoder and doing a DE analysis on the filtered transcriptome? I have found, in my own datasets, that this can help remove a lot of noise, reduce false discovery rate correction, and identify some more genes with BLAST hits.

Additional comments

Overall, I think this is a well written paper with interesting results! I would love to see the additional molecular work to identify splice variants. Failing that, I would also be happy to see this paper published if the wording of the first hypothesis was altered and discussion and conclusions around the expression analysis of EF1A were toned down.

---

## Round 0.2 · accepted · Accept

Thank you very much for your careful attention to the comments of the reviewers and provided a clear rationale for your responses. Although I think I understand the intent of the following sentence in the discussion "Nevertheless the effect of our temperature stress trial suggests heritable variation in how individuals respond to heat stress" I would encourage you to clarity it with an additional comment.